# Evaluating Out-of-Distribution Performance on Document Image Classifiers

**Stefan Larson**[*1,3]     **Gordon Lim**[2]     **Yutong Ai**[2]     **David Kuang**[2]     **Kevin Leach**[3]

[1]DryvIQ
Ann Arbor, MI, USA

[2]University of Michigan
Ann Arbor, MI, USA

[3]Vanderbilt University
Nashville, TN, USA

## Abstract

The ability of a document classifier to handle inputs that are drawn from a distribution different from the training distribution is crucial for robust deployment and generalizability. The RVL-CDIP corpus [18] is the *de facto* standard benchmark for document classification, yet to our knowledge all studies that use this corpus do not include evaluation on *out-of-distribution* documents. In this paper, we curate and release a new out-of-distribution benchmark for evaluating out-of-distribution performance for document classifiers. Our new out-of-distribution benchmark consists of two types of documents: those that are not part of any of the 16 in-domain RVL-CDIP categories (RVL-CDIP-*O*), and those that are one of the 16 in-domain categories yet are drawn from a distribution different from that of the original RVL-CDIP dataset (RVL-CDIP-*N*). While prior work on document classification for in-domain RVL-CDIP documents reports high accuracy scores, we find that these models exhibit accuracy drops of between roughly 15-30% on our new out-of-domain RVL-CDIP-*N* benchmark, and further struggle to distinguish between in-domain RVL-CDIP-*N* and out-of-domain RVL-CDIP-*O* inputs. Our new benchmark provides researchers with a valuable new resource for analyzing out-of-distribution performance on document classifiers. Our new out-of-distribution data can be found at `github.com/gxlarson/rvl-cdip-ood`.

## 1 Introduction

The task of automated document classification has wide-ranging use cases, especially in industry where it can be used to apply labels to massive amounts of documents. In many scenarios, a model is trained on an initial training set, which is drawn from a particular distribution which may have certain characteristics (e.g., all documents are in the English language, are from a particular industry or time period, etc.). It is often desirable for the model to be able to *generalize* to valid inputs that are *out-of-distribution* and exhibit different characteristics than those seen in the *in-distribution* training data. At the same time, it is desirable for a model operating in unconstrained input spaces to be able to identify inputs that are not part of the target *label* set as *out-of-domain* to minimize false-positive predictions.

The RVL-CDIP corpus has emerged as the *de facto* benchmark dataset for document classification. This single-label classification dataset covers 16 document categories (e.g., `resume`, `invoice`, `letter`, `memo`, etc.), and consists of samples taken from a large collection of documents made public as a result of litigation and settlement agreements involving the tobacco industry. As such, most of

---

[*]Corresponding email: `slarson@dryviq.com` (alternate email: `stefan.dataset@gmail.com`).

36th Conference on Neural Information Processing Systems (NeurIPS 2022) Track on Datasets and Benchmarks.

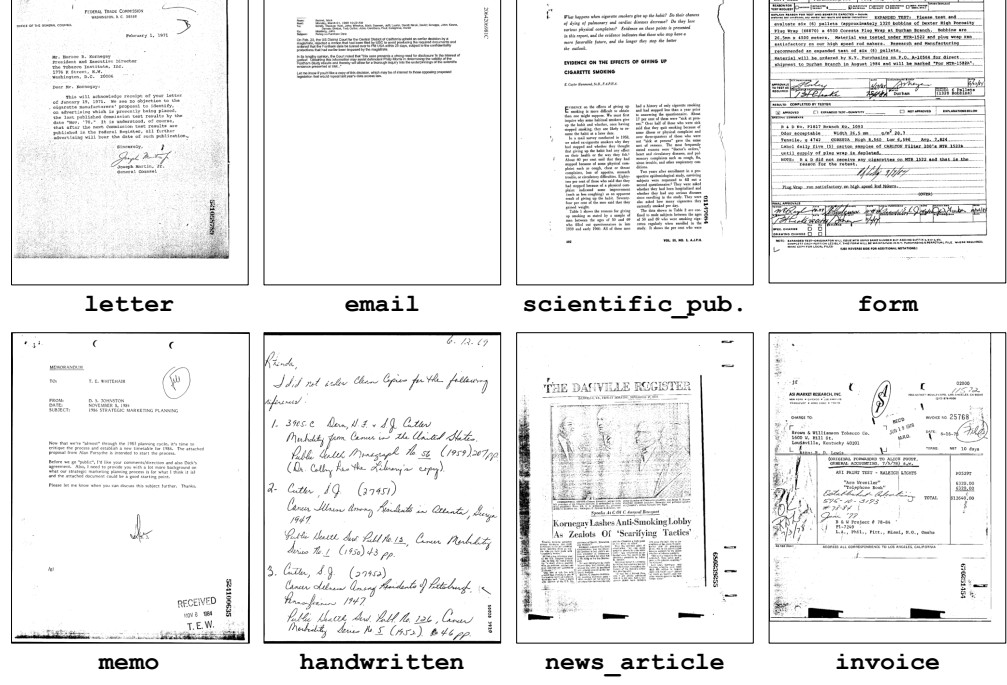

| letter | email | scientific_pub. | form |
| --- | --- | --- | --- |
| memo | handwritten | news_article | invoice |

Figure 1: Samples from the RVL-CDIP corpus.

the RVL-CDIP documents are related to the American tobacco industry, and were created before 2006. Samples of RVL-CDIP are displayed in Figure 1.

Many contemporary models achieve high classification accuracy scores on the RVL-CDIP classification benchmark (e.g., [34, 47, 57, 58]). Despite this, there has been limited analysis of these models' ability to handle out-of-distribution inputs. In this paper, we address this gap by creating a new out-of-distribution (OOD) test set intended to be used in conjunction with RVL-CDIP. We first identify two types of OOD inputs: (1) in-domain documents that fall within one of the 16 RVL-CDIP categories, yet were sampled from a different distribution than RVL-CDIP (for example, born-digital documents from the 2010s and 2020s, and those not from the tobacco industry); and (2) out-of-domain documents that do not fall within one of the 16 RVL-CDIP categories. Then, we evaluate several document classifiers trained on RVL-CDIP and tested on our new OOD data. We find that model performance on both types of OOD data is substantially lower than on the in-distribution RVL-CDIP test set, indicating that while models may perform well on RVL-CDIP, they (1) struggle to generalize to other distributions and (2) struggle on the task of out-of-domain document classification. Our hope is that with our new OOD data, researchers investigating document classification will also begin to include OOD performance as part of their analyses.

## 2 Related Work

### 2.1 RVL-CDIP Document Classification

The RVL-CDIP corpus [18] is widely used for benchmarking the performance of document classifiers. This corpus consists of images of scanned documents, and most early efforts on the RVL-CDIP classification task used convolutional neural networks (CNNs), including [1, 10, 18, 27, 52]. Later work incorporated the textual modality along with visual features (e.g., [3, 4, 11, 15]). The recent advent of transformers has seen researchers employ this architecture to the classification task. This includes models that solely use image features (e.g., [12, 25, 34]) as well as models that incorporate page layout and optical character recognition (OCR) features (e.g., [2, 6, 13, 16, 33, 37, 44, 46, 47, 48, 54, 57, 58]). Despite the prominence of the RVL-CDIP corpus in evaluating document classifiers, there is no published OOD corpus for benchmarking models on out-of-distribution performance.

## 2.2 Out-of-Distribution Performance Benchmarking

Prior work has developed benchmarks for evaluating out-of-distribution performance for other tasks. In general, these benchmarks belong to one of two categories: (1) out-of-domain or out-of-scope benchmarks, where—in the case of classification—dedicated evaluation data is included that does not belong to any of the in-domain target labels; and (2) distribution shift benchmarks, where evaluation data is different than the training distribution yet still belongs to the in-domain target label set. We discuss relevant work in each category.

*Out-of-scope* or *out-of-domain* evaluation benchmarks are used to measure a model's ability to differentiate between in-domain and out-of-domain inputs. In this setting, a model is typically trained on the in-domain data, and then evaluated on both in- and out-of-domain data (where ideally the model can correctly classify in-domain data with high accuracy while also being able to discriminate between in- and out-of-domain data, which could be done using a confidence score). Out-of-domain benchmarks include those for text classification (e.g., intent classification in dialog systems [30]) and image classification and object detection in the presence of *natural adversarial* inputs that mislead classifiers (e.g., ImageNet-O [23] and its COCO counterpart [31]). In this paper, we present new out-of-domain evaluation data for the RVL-CDIP benchmark, which we call RVL-CDIP-*O*.

*Distribution shift* evaluation benchmarks are crafted to measure a model's ability to generalize to "unseen" inputs, as well as to measure how robust a model is to perturbations and/or corruptions (e.g., noise). Here, "unseen" typically means data that is different from the training distribution, yet still belongs to the label-set of the training dataset. For image classification, the "seen" training distribution may be photographic pictures of images but the "unseen" data distribution may be different "views" or modalities of these images, like hand-drawn sketches or clip art representations of the image categories (e.g., DomainNet [45] and Office-Home [53]).

Other domain-shift benchmarks include the WILDSbenchmark [26], which itself is a collection of 10 mostly image and text datasets for evaluating out-of-distribution performance. For instance, WILDS includes the Camelyon17 histopathology dataset [7], where the task is image classification but where the test set consists of data collected from a hospital different from that of the training set. Other prior work (e.g., [24, 35, 36]) evaluates out-of-distribution detection for image classifiers trained on CIFAR-10/100 by testing on various other image datasets, but does not account for the problem of discriminating between out-of-distribution inputs that are out-of-domain versus in-domain.

Lastly, datasets that measure a model's robustness to corruptions include ImageNet-C [20], which uses data augmentation strategies to generate noisy versions of ImageNet [14] images. We note that other data augmentation tools and pipelines have been used to generate perturbations and corruptions of in-distribution data, including [40, 43, 49, 39] for image and text data. In this paper, we present new evaluation data for benchmarking both of the out-of-distribution tasks described in this section. Our new benchmark data enables researchers to analyze document classifier performance on out-of-domain inputs as well as in-domain data from a "shifted" input distribution.

## 3 Datasets

In this section, we present our new evaluation data for benchmarking out-of-distribution performance on document classification models. We first discuss the RVL-CDIP dataset as it is relevant in constructing our new out-of-distribution data. We then discuss our new out-of-distribution benchmark.

## 3.1 RVL-CDIP

The RVL-CDIP corpus consists of grayscale images of scanned documents from the IIT-CDIP collection [32], a large repository of publicly-available documents that were released as part of litigation against several tobacco-related companies and organizations. As such, all documents in the RVL-CDIP corpus are tobacco-related. The corpus consists of 16 categories: `advertisement`, `budget`, `email`, `file_folder`, `form`, `handwritten`, `invoice`, `letter`, `memo`, `news_article`, `presentation`, `questionnaire`, `resume`, `scientific_publication`, `scientific_report`, and `specification`. Samples from RVL-CDIP are shown in Figure 1. We note that many of the RVL-CDIP samples contain visual noise introduced by scanners due to having been scanned from the physical world using old or noisy document scanners. Each RVL-CDIP category has 20,000 training samples (320,000 total training samples), and there is a total of 40,000 validation and 40,000

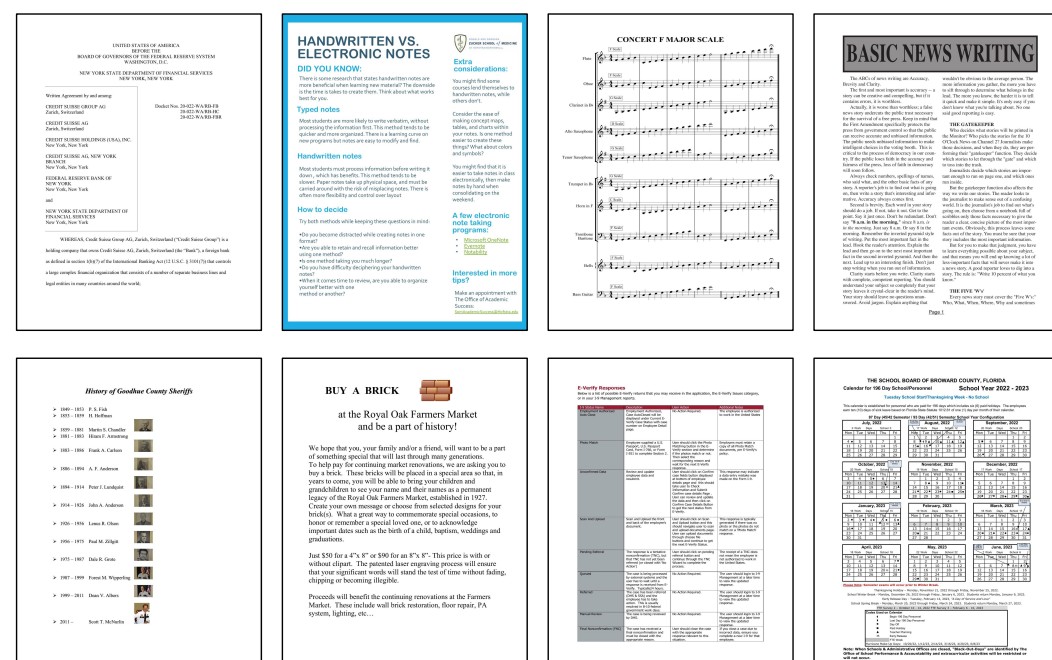

Figure 2: Example out-of-distribution document images from our RVL-CDIP-*O* set, prior to grayscale transformation.

test documents across all categories. All documents from this dataset are from the year 2006 or earlier, as 2006 was the year that the IIT-CDIP collection was released.

## 3.2 New Out-of-Distribution Data

Our new out-of-distribution benchmark consists of two subsets of data: document images that (a) do not belong to any of the 16 in-domain RVL-CDIP categories (we call this subset RVL-CDIP-*O*); and (b) belong to one of the 16 RVL-CDIP categories yet are not from IIT-CDIP or tobacco-related (RVL-CDIP-*N*). Examples from RVL-CDIP-*O* are shown in Figure 2, while examples from RVL-CDIP-*N* are shown in Figure 3. The samples in Figure 2 show several types of documents that are not included in the 16 RVL-CDIP categories, including—but not limited to—court documents, guides, music sheets, timelines, and schedules.

Our out-of-distribution documents were collected from two internet sources: (1) Google and Bing web searches; and (2) the public DocumentCloud [2] repository. The DocumentCloud

Table 1: Per-category counts for RVL-CDIP-*N*.

| Category | Count |
|---|---|
| budget | 58 |
| email | 33 |
| form | 70 |
| handwritten | 176 |
| invoice | 57 |
| letter | 152 |
| memo | 47 |
| news_article | 86 |
| questionnaire | 39 |
| resume | 184 |
| scientific_pub. | 39 |
| specification | 61 |
| Total | 1,002 |

repository contains a large number of government, legal, or public service related documents that were made available through serviced Freedom of Information Act (FOIA) requests. To collect the new OOD data, the authors of this paper first familiarized themselves with the 16 in-domain RVL-CDIP categories by reviewing numerous samples from each RVL-CDIP category. While reviewing samples from RVL-CDIP, we noticed that some could be seen as having multiple labels—for instance, a handwritten letter could technically be both handwritten and letter. However, we found that

---

[2] https://www.documentcloud.org

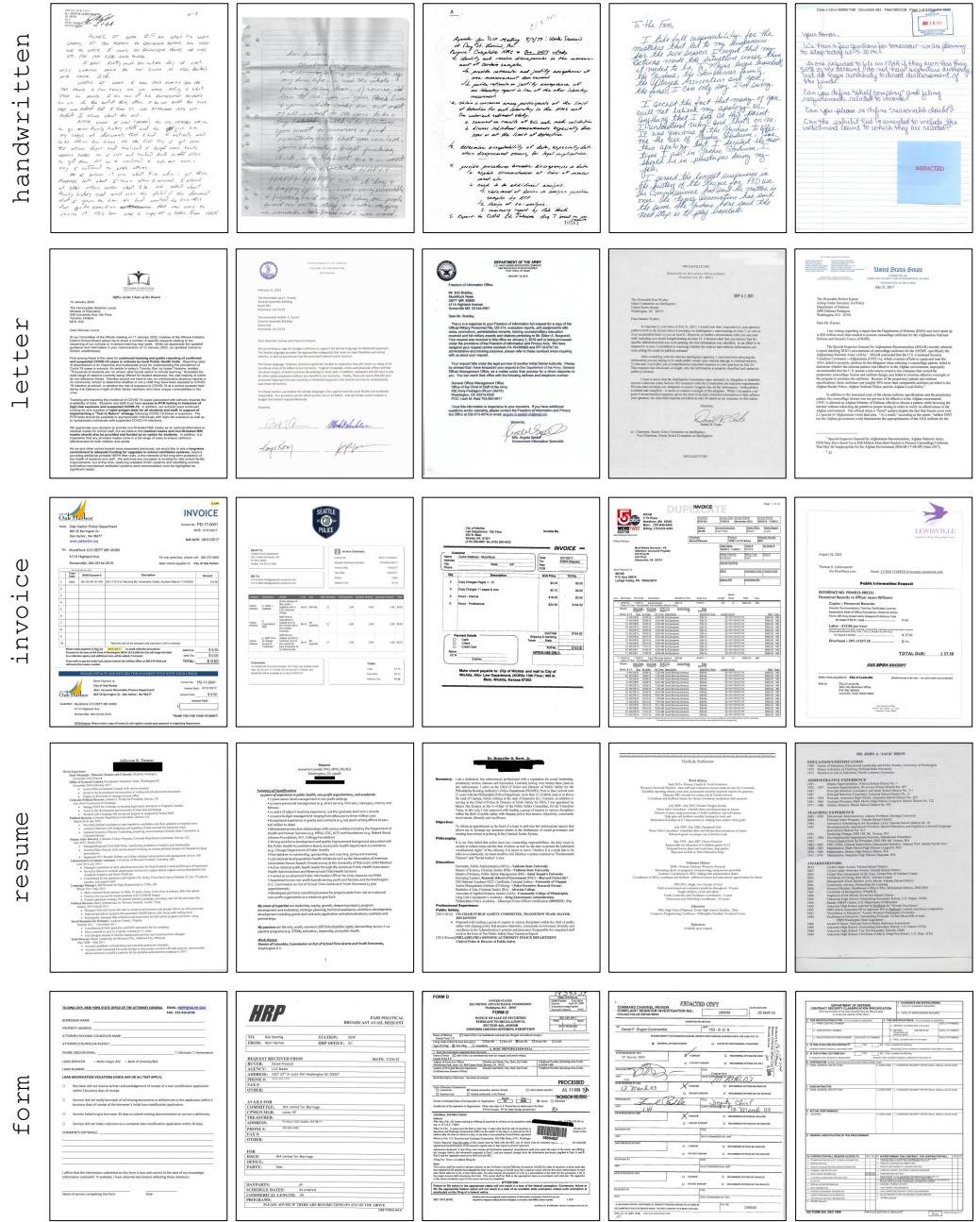

Figure 3: Samples from the RVL-CDIP-*N* evaluation set, prior to grayscale transformation.

RVL-CDIP is consistent in labelling such cases as with a single label (e.g., handwritten letters are labeled as handwritten). The annotations we applied to our new RVL-CDIP-*N* adopt such rules from RVL-CDIP.

To collect RVL-CDIP-*O* data, the authors searched for PDF documents that were not one of the 16 RVL-CDIP categories. Similarly for RVL-CDIP-*N*, the authors searched for PDF documents that *were* one of the 16 RVL-CDIP categories. While original descriptions of the RVL-CDIP categories are unavailable, we sought to include only clear-cut examples of these categories in our RVL-CDIP-*N* set; samples from each RVL-CDIP-*N* category are juxtaposed with original RVL-CDIP samples in Figures 20 – 43 (in Appendix B). Using our search method, we were unable to find sufficient

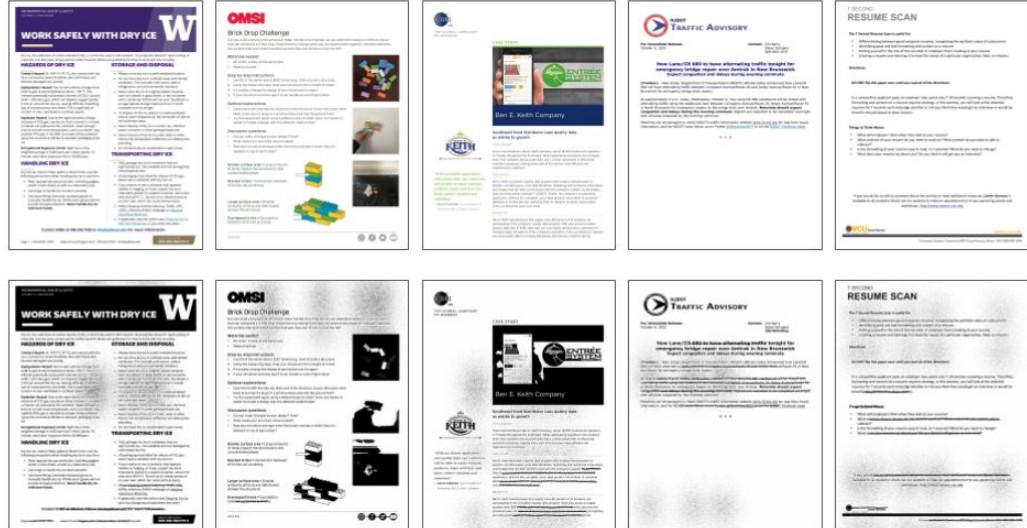

Figure 4: Examples of document images from the RVL-CDIP-*O* set (top row) with corresponding Augraphy-augmented grayscale images (bottom row).

documents for a few of the in-domain categories like `file_folder`, as that category seems to be quite specific to the RVL-CDIP dataset. Counts of the in-domain categories that we collected for RVL-CDIP-*N* are shown in Table 1.

Each raw document may have consisted of multiple pages, but to maintain consistency between our out-of-distribution data and the original RVL-CDIP data, we only use the first page of each document, which we convert to a grayscale image. Following the RVL-CDIP dataset's convention, we save each image so that the maximum dimension (i.e., height or width) is 1000 pixels, and like RVL-CDIP we save the OOD images to TIFF files. As summarized in Table 2, there are a total of 4,417 out-of-distribution document images. We estimate that roughly 65% of documents that we collected from DocumentCloud are "born digital" (i.e., they are "clean" and not scanned versions of physical paper documents), while around 95% of documents that we collected via web search are born digital. For this reason we also conduct a side experiment with the Augraphy tool [39] to add scanner-like noise to our out-of-distribution data as a way to test whether models trained on RVL-CDIP are overfit to scanner-like noise. Examples of our data post-Augraphy are shown in Figure 4. Our final, published data for both RVL-CDIP-*N* and RVL-CDIP-*O* do not include Augraphy augmentations.

Our new data is out-of-distribution with respect to the RVL-CDIP corpus in several ways: (1) a substantial portion of our new data is "born-digital", whereas the overwhelming majority of RVL-CDIP is scanned physical documents; (2) a substantial portion of our data was created post-2006, with a large amount having been created within the past 10 years; (3) documents from our dataset are almost exclusively from industries and topics other than tobacco-related ones. As such, our new out-of-distribution data poses a challenge to models trained on RVL-CDIP, as models evaluated on our new data must be able to generalize to new distributions (RVL-CDIP-*N*), while also being able to distinguish between in- and out-of-domain inputs (RVL-CDIP-*O*).

Table 2: Out-of-distribution dataset count statistics. DC and WS refer to Document-Cloud and Web Search, respectively.

|  | DC | WS | Total |
| --- | --- | --- | --- |
| RVL-CDIP-*N* | 877 | 125 | 1,002 |
| RVL-CDIP-*O* | 1,382 | 2,033 | 3,415 |
| Total | 2,259 | 2,158 | 4,417 |

## 4   Experiments

Recall that our proposed dataset is intended to reveal deficiencies in recognizing out-of-distribution data in document classification tasks. We demonstrate our dataset's viability by (1) training several classifiers on the full RVL-CDIP training set, then (2) evaluating the performance of each model on

our new out-of-distribution data. We discuss the models and metrics used in our experiments, and then discuss the results obtained.

## 4.1 Models

We trained several image-based classifiers on the full RVL-CDIP training set, including convolutional neural network (CNN) architectures and image transformer architectures. We consider the following CNNs: VGG-16, ResNet-50, GoogLeNet, and AlexNet (each uses the architecture from [50, 19, 51, 28], respectively). All of these CNN models were originally trained and evaluated on the RVL-CDIP corpus by [1]. Our analysis uses PyTorch [41] implementations of these models. In addition, we evaluate the LayoutLMv2-base model architecture [58], which uses a language model trained on the locations and sizes of words in documents. As the original RVL-CDIP corpus does not provide textual information, we use the Tesseract [3] OCR engine to extract words and their locations in each document image. Our analysis uses the Hugging Face [56] implementation of LayoutLMv2.

We also evaluate the image-only transformer-based DiT model [34], which itself is a version of the BEiT image transformer [8]. Recent work has shown that image transformers often outperform CNN models, and prior work has also found that transformer models outperform other model architectures in terms of out-of-distribution performance (e.g., [9, 22, 38, 42]; on the contrary, cf. [5, 55] for opposing viewpoints). Unlike the other models analyzed here, we use the pre-trained DiT model hosted on Hugging Face instead of training the model ourselves. All models have an input image resolution of 224x224 pixels.

## 4.2 Evaluation Metrics

**RVL-CDIP-*N*.**  We measure each model's raw classification accuracy on our RVL-CDIP-*N* dataset, as RVL-CDIP-*N* contains document images that *do* belong to one of the 16 in-domain RVL-CDIP document categories. Accuracy can be measured by aggregating all predictions together (micro accuracy), or by averaging the within-category accuracy scores (macro accuracy); we report both.

**RVL-CDIP-*O*.**  The documents from RVL-CDIP-*O* do not belong to any of the 16 RVL-CDIP categories, and we do not train the models on a $17^{th}$ catch-all `unknown` category for out-of-domain documents. Therefore, we benchmark two decision functions that use the model's output logits as a way to arbitrate between in- and out-of-domain predictions. The first, termed Maximum Softmax Probability (MSP) obtains confidence scores from applying the softmax function on the logits (as done in e.g. [21, 29, 30]). If the model $f$ applied to a document image $x$ produces logits $f(x) = z$, then MSP applies the softmax function to obtain confidence scores scaled between 0 and 1 for each in-domain category. That is, the $i^{th}$ document category has a confidence score of

$$s(z)_i = \frac{e^{z_i}}{\sum_{j=1}^{K} e^{z_j}},$$

where $K$ is the number of target label categories (in our case, 16). An alternative confidence function that has been shown to perform better on out-of-distribution detection tasks than MSP is the Energy score [36], defined as

$$E(z;T) = T \cdot log \sum_{j=1}^{K} e^{z_j/T}$$

where $T$ is a temperature parameter. Note that the Energy score produces a single confidence score rather than $K$ scores.

An ideal classifier will assign high confidence to an in-domain document, and lower confidence to an out-of-domain document. One straightforward way to do this is to use a threshold on the confidence score, so that we assign an `unknown` label to documents that have a confidence score lower than *t*. That is, the decision rule is

$$\text{decision rule} = \begin{cases} \text{in-domain,} & \text{if } \max(score(z)) \geq t \\ \text{out-of-domain,} & \text{if } \max(score(z)) < t. \end{cases}$$

---

[3] https://github.com/tesseract-ocr/tesseract

Table 3: Accuracy scores on RVL-CDIP compared to RVL-CDIP-*N*.

| Model | RVL-CDIP (reported) | RVL-CDIP (achieved) | RVL-CDIP-*N* micro | RVL-CDIP-*N* micro w/Aug. | RVL-CDIP-*N* macro | RVL-CDIP-*N* macro w/Aug. |
|---|---|---|---|---|---|---|
| VGG-16 | 0.910 | 0.905 | 0.668 | 0.650 | 0.691 | 0.678 |
| ResNet-50 | 0.911 | 0.900 | 0.598 | 0.606 | 0.615 | 0.621 |
| GoogLeNet | 0.884 | 0.871 | 0.602 | 0.594 | 0.613 | 0.591 |
| AlexNet | 0.900 | 0.885 | 0.567 | 0.559 | 0.571 | 0.551 |
| LayoutLMv2 | 0.953 | 0.887 | 0.556 | 0.563 | 0.600 | 0.606 |
| DiT-base | 0.921 | 0.933 | 0.786 | 0.706 | 0.805 | 0.725 |

Note that *t* could be global or specific to each document category (as noted in [17]). Our goal is then to measure how well the decision rule is able to correctly identify in- versus out-of-domain inputs (i.e., this is a binary classification problem). While we could set a threshold (or thresholds) in our analysis, we instead opt for a threshold-free evaluation that uses Area Under the Reciever Operating Characteristic curve (AUC) scores to measure the separability of confidence scores between in- and out-of-domain documents. The AUC score ranges between 0.5 and 1.0, where higher AUC scores mean a model is more able to assign lower confidence to out-of-domain documents and higher confidence to in-domain documents, while a low AUC score of 0.5 means the distributions of confidence scores for in- and out-of-domain documents are roughly the same.

In our analysis, we compute the AUC between the confidence scores on RVL-CDIP test set and RVL-CDIP-*O*, as well as RVL-CDIP-*N* and RVL-CDIP-*O* (the latter pair being a more challenging task, as both RVL-CDIP-*N* and RVL-CDIP-*O* are out-of-distribution). Like accuracy, we report both macro and micro AUC scores for each model, which is important for the RVL-CDIP-*O* data because the distribution of confidence scores may vary between predicted label categories. Macro AUC is computed by computing the AUC for each document category, where the samples used in the computation of each category's AUC score are those samples where the model predicted the sample as that category. Then, each category's AUC is averaged to yield the macro AUC score. We also compute the false-positive rate at 95% true positive rate (FPR95), but these results are presented in Appendix A.3. To compute FPR95, a threshold *t* is selected so that the true-positive rate is 95%, and then compute the false-positive prediction rate for out-of-domain documents at this threshold. FPR95 is a lower-is-better metric.

## 4.3 Results

Table 3 displays accuracy on the RVL-CDIP test set as reported by prior work, as well as the scores achieved by us. Table 3 also displays accuracy scores on our new RVL-CDIP-*N* test set. First, we note that we achieve in-domain RVL-CDIP test scores that are close to the reported scores, with the exception of LayoutLMv2. One difference between our setup and the LayoutLMv2 setup used in [58] is the OCR engine used: we used the open-source Tesseract library whereas [58] used a proprietary Microsoft Azure OCR model. In other words, OCR quality may have impacted our trained LayoutLMv2 model.

### 4.3.1 Performance on RVL-CDIP-*N*

Table 3 shows that the models appear to underperform on the out-of-distribution document images from the new RVL-CDIP-*N* set, with micro accuracy scores ranging in the mid 0.500s to 0.786 (in the case of DiT). These accuracy scores are substantially lower than the scores on the in-distribution RVL-CDIP test set. Even the top-performing DiT model sees an micro accuracy drop of roughly 15 percentage points. Examples of errors made by the DiT model on RVL-CDIP-*N* can be seen in Figure 7 (bottom row). Using the Augraphy augmentation tool to introduce scanner-like noise to the RVL-CDIP-*N* test images does little to impact accuracy, and we observe no clear trend under this setting. In summary, we observe that models trained on RVL-CDIP exhibit substantially lower accuracy scores on our out-of-distribution RVL-CDIP-*N* evaluation set than on the original RVL-CDIP test set, indicating that these models do not generalize well to new data.

### 4.3.2 Performance on RVL-CDIP-*O*

Table 4 charts AUC scores on the task of out-of-domain detection for models trained on RVL-CDIP but tested on the out-of-domain data from our new RVL-CDIP-*O* data. This table shows AUC scores

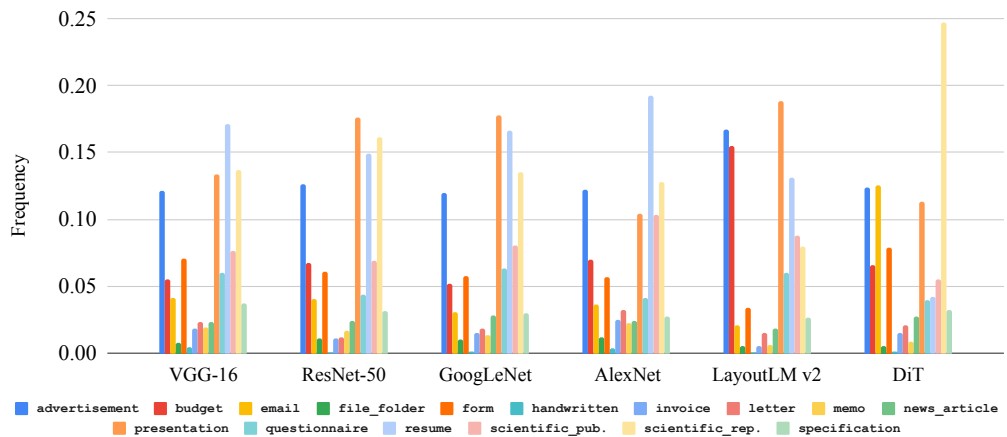

Figure 5: Distributions of predicted labels for each supervised model on the out-of-domain RVL-CDIP-*O* data.

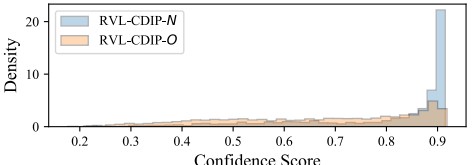 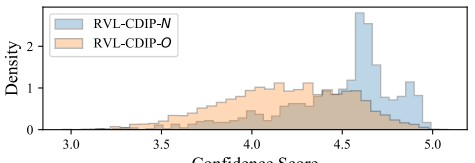

Figure 6: Distribution of confidence scores using MSP (left) and Energy (right) for DiT on RVL-CDIP-*N* and RVL-CDIP-*O*.

for model confidence score output on RVL-CDIP versus RVL-CDIP-*O* (*T-O*) as well as RVL-CDIP-*N* versus RVL-CDIP-*O* (*N-O*); both of these settings explore a model's ability to differentiate between in-domain (RVL-CDIP and RVL-CDIP-*N*) versus out-of-domain (RVL-CDIP-*O*) data, but the *N-O* setting is more challenging, as both RVL-CDIP-*N* and RVL-CDIP-*O* are out-of-distribution with respect to RVL-CDIP. We see that most models achieve AUC scores on RVL-CDIP versus RVL-CDIP-*O* (*T-O*) within the mid-to-high 0.800s or low 0.900s for both MSP and Energy confidence scoring methods. In contrast, models perform much worse on the task of discriminating between RVL-CDIP-*N* and RVL-CDIP-*O* (*N-O*)—with DiT dropping in performance by 16.6 points for MSP and 13.5 points for Energy—indicating that the document classification models have much more difficulty detecting out-of-domain documents in the presence of in-domain but out-of-distribution documents.

Figure 6 casts light as to why, as we see that there is a large amount of overlap between confidence scores for RVL-CDIP-*N* and RVL-CDIP-*O* documents for both MSP and Energy approaches using the DiT model. Similar plots for the other models can be found in Figures 8–19 in Appendix A.4, where we also plot model performance against confidence score threshold. To correctly identify more out-of-domain documents—*ceteris paribus*—we must naturally increase confidence score thresholds. Thus we observe in Figures 8–19 that classification accuracy on RVL-CDIP-*N* naturally decreases as we increase the confidence threshold, as more in-domain documents are mis-detected as out-of-domain at higher confidence thresholds.

Table 4: AUC scores on RVL-CDIP and RVL-CDIP-*O* (*T-O*) and on RVL-CDIP-*N* and RVL-CDIP-*O* (*N-O*) for MSP and Energy confidence scoring methods.

| Model | *T-O* MSP | *T-O* Energy | *N-O* MSP | *N-O* Energy |
|---|---|---|---|---|
| VGG-16 | 0.881 | 0.923 | 0.649 | 0.648 |
| ResNet-50 | 0.849 | 0.844 | 0.581 | 0.554 |
| GoogLeNet | 0.838 | 0.847 | 0.592 | 0.587 |
| AlexNet | 0.871 | 0.909 | 0.620 | 0.646 |
| LayoutLMv2 | 0.842 | 0.849 | 0.620 | 0.662 |
| DiT-base | 0.894 | 0.888 | 0.728 | 0.753 |

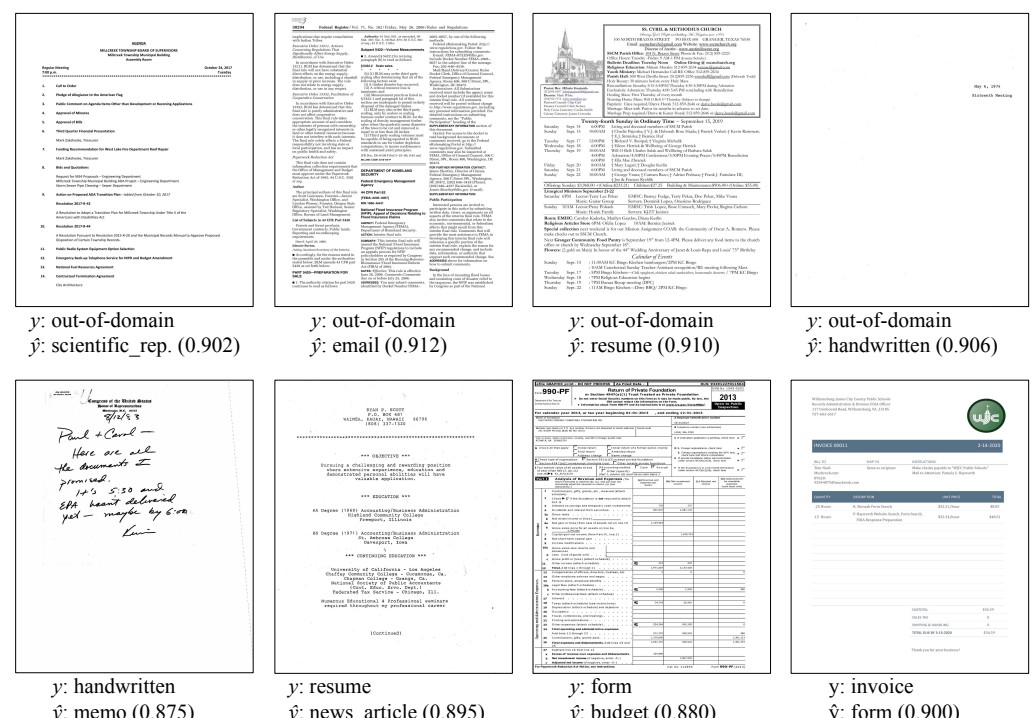

Figure 7: DiT model mis-predictions. Top row: errors on the RVL-CDIP-*O* set. Bottom row: errors on the RVL-CDIP-*N* set. True labels are indicated by *y*, while predicted labels are indicated with *ŷ* (and confidence scores in parentheses).

We delve further into the predictions made by these models by investigating the distribution of labels for the predictions on the RVL-CDIP-*O* test data: Figure 5 plots the predicted label distribution for the data from RVL-CDIP-*O*. We observe several trends: most of the supervised models have relatively higher amounts of predictions for `advertisement`, `presentation`, `scientific_publication`, and `resume` labels, perhaps due to these categories having similar visual features to much of the RVL-CDIP-*O* data. Examples of high-confidence predictions on RVL-CDIP-*O* can be seen in Figure 7 (top row). Overall, our findings of model performance on differentiating between in- and out-of-domain inputs show room for improvement in the case where both in- and out-of-domain documents are out-of-distribution with respect to the original RVL-CDIP training set.

## 5   Conclusion

In this paper, we introduce new data for evaluating out-of-distribution performance on document classification models trained on the popular RVL-CDIP benchmark. Our new out-of-distribution data is composed of two types: RVL-CDIP-*N*, or data that *do belong* to one of the 16 in-domain document categories in RVL-CDIP yet were obtained from a different distribution than the original RVL-CDIP corpus; and RVL-CDIP-*O*, or out-of-domain data that do *not* belong to one of the 16 RVL-CDIP label categories. We find that while models trained on RVL-CDIP perform well on the in-distribution RVL-CDIP test data, they struggle comparatively on the out-of-distribution RVL-CDIP-*N* data as well as at differentiating between the out-of-domain RVL-CDIP-*O* and in-domain RVL-CDIP-*N* data.

Analyzing model performance on out-of-distribution inputs is an important yet understudied problem, but is a hurdle for developing robust, generalizable models. This seems to be the case in the document classification field, where most analyses rely only on in-distribution and in-domain tests performed on RVL-CDIP. Our hope is that our new out of distribution data—RVL-CDIP-*N* and RVL-CDIP-*O*—provide researchers with a valuable resource for analyzing out-of-distribution performance on document classifiers.

## Acknowledgments

We thank the University of Michigan's Undergraduate Research Opportunity Program (UROP) for supporting Gordon and Yutong. We also thank the University of Michigan's MRADS program for supporting Gordon. We thank Ann Arbor SPARK and the Michigan STEM Forward program for partial support of David. We thank Brian Yang for his help in early phases of data collection, and the anonymous NeurIPS reviewers for their feedback.

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
