# OpenReview forum: "Evaluating Out-of-Distribution Performance on Document Image Classifiers"
_NeurIPS.cc/2022/Track/Datasets_and_Benchmarks — NeurIPS 2022 Datasets and Benchmarks _

### Official Review · Reviewer_cEE9 · 2022-07-22
**A New Out-of-Distribution Image Benchmark But Limited Novelty**

**Rating:** 6
**Confidence:** 4

**Strengths:**

1. The dataset, to my knowledge, is the first out-of-distribution dataset for document image classification.
2. Experiments show the performance gap between the original corpus and the new dataset.

**Weaknesses:**

1. The proposed dataset is not novel enough, which is obtained from the existing corpus.
3. The benchmark is an image dataset in nature, and it is unclear how it compares with other OOD image datasets.
2. The documentation of the benchmark is poor.


**Additional Feedback:**

I consider raising the score if the authors can provide a satisfactory rebuttal to my concerns.

**Clarity:**

The paper is overall well-organized. clearly written, and easy to follow. But some acronyms should be introduced with their full names like OCR in Section 2.1

**Correctness:**

One major concern is the significance of the proposed benchmark. The authors claim that this is the first benchmark for the OOD document image benchmark. However, there exist other OOD image benchmarks such as all datasets used from the generalized ODIN paper.
Hsu, Yen-Chang, et al. "Generalized odin: Detecting out-of-distribution image without learning from out-of-distribution data." Proceedings of the IEEE/CVF Conference on Computer Vision and Pattern Recognition. 2020.
It is much better if authors can discuss how their benchmark is different from them.

**Documentation:**

Another major concern is poor documentation. The authors only upload their datasets without any explanation. It is much better if the authors can provide a readme file to illustrate how to use this benchmark, including how to generate the training set and the test set. The authors also highly recommended preparing a demo, in order to help readers get familiar with their benchmark.

**Ethics:**

No ethical issue is found.

**Relation To Prior Work:**

The relation to previous work is well illustrated and easy to understand.

**Summary And Contributions:**

This paper proposes a new image benchmark based on the RVL-CDIP corpus. The benchmark consists of two datasets: RVL-CDIP-N contains all documents which do not fall in any 16 domains of the RVL-CDIP corpus and RVL-CDIP-O contains all documents which fit in 16 domains of the RCL-CDIP corpus but comes from a different distribution. Experiments demonstrate that existing approaches perform 15%-30% lower on the new RVL-CDIP-N datasets than the original corpus due to the out-of-distribution.  The new benchmark is released.

---

> ### Author Response · Authors · 2022-08-25
> **Response to Reviewer cEE9**
>
> Thank you for your review.
>
> - _The proposed dataset is not novel enough, which is obtained from the existing corpus._
>
> Our paper introduces a new dataset of two types of OOD data for evaluating OOD performance on document classifiers trained on RVL-CDIP. The RVL-CDIP corpus is the de facto standard benchmark for document classification, but currently there are no studies that evaluate performance of document classifiers on OOD inputs. Our new dataset fills this gap and provides researchers with the opportunity to benchmark their models on OOD inputs. Our dataset is not obtained from any existing corpus. However, there are many other datasets that are designed to be companion datasets that have had a wide impact, like ImageNet-C (https://arxiv.org/pdf/1903.12261.pdf; published in ICLR) and ImageNet-O (https://arxiv.org/pdf/1907.07174.pdf; published in CVPR).
>
> - _The benchmark is an image dataset in nature, and it is unclear how it compares with other OOD image datasets._ and _ there exist other OOD image benchmarks such as all datasets used from the generalized ODIN paper._
>
> Our dataset includes two types of OOD data: data that belongs to the in-domain RVL-CDIP categories (we call this set RVL-CDIP-N), as well as data that does not belong to any of the 16 RVL-CDIP categories (we call this set RVL-CDIP-O). One major difference between our work and the datasets/evaluation used in the Generalized ODIN paper is that our work includes the former type (RVL-CDIP-N) of OOD data, while the Generalized ODIN paper/datasets do not. This allows us to test OOD performance more broadly. The Generalized ODIN paper/datasets are more akin to our RVL-CDIP-O set, but the Generalized ODIN datasets are general image classification datasets, while ours is specific to document classification, which is a field that has seen new model types specific to the task of document classification (e.g. DiT, LayoutLMv2, etc.). We have added additional text to the Related Work section of the paper to discuss this.
>
> - _The documentation of the benchmark is poor._
>
> We have updated the dataset repository to include more documentation and evaluation code.

---

> > ### Comment · Reviewer_cEE9 · 2022-08-25
> > **Thank you for the classification**
> >
> > I appreciate efforts authors made to improve their paper.  The rebuttal addresses my major concerns, and therefore raise my score to weak accept.

---

### Official Review · Reviewer_1zH1 · 2022-07-26
**Marginal Contribution**

**Rating:** 6
**Confidence:** 5
**Correctness:** see previous section.
**Clarity:** contains several typos and mistakes.

**Strengths:**

The problems of both distributional shift and out-of-distribution are relevant to real-world applications of machine learning models. The collected dataset seems to be manually annotated (but not explicitly stated in the main text).

**Weaknesses:**

1. The writing is not careful. e.g. in L9 of abstract, the RVL-CDIP-N is defined as ood, which is a mistake.
2. P7 line10 from bottom. "logits (typically done using the softmax function)". The authors confuses logits with probabilities.
3. P7 line3 from bottom. "report both macro and micro AUC". It is unclear how the "macro AUC" is computed considering that the decision rule of ood need to take the maximum over all classes.
4. P5 line3 from bottom. It is not wise to pollute the raw data with simple augmentations, because (a) augmentations can be also used in training, which makes the perturbation of testing images pointless (b) augmentation creates simple artifacts that can be exploited by ood algorithms, making the dataset less useful.
5. only one ood (MSP) algorithm is used in ood evaluation, so the conclusion "struggle on the task of out-of-domain document classification" is less convincing.

**Additional Feedback:**

/

**Documentation:**

Not sure if it is manually labelled, or what the quality of labelling is.

**Ethics:**

The dataset contains human faces.

**Relation To Prior Work:**

yes

**Summary And Contributions:**

The paper collects two sets of data RVL-CDIP-N and RVL-CDIP-O, which serves as an domain shifted and out-of-distribution testing dataset for the RVL-CDIP dataset. Several models are evaluated on the new datasets and it is observed that models do not perform well on neither domain shifted or ood datasets.

---

> ### Author Response · Authors · 2022-08-25
> **Response to Reviewer 1zH1**
>
> - _The writing is not careful. e.g. in L9 of abstract, the RVL-CDIP-N is defined as ood, which is a mistake._
>
> This typo has been corrected in the updated version of the paper.
>
> - _P7 line10 from bottom. "logits (typically done using the softmax function)". The authors confuses logits with probabilities._
>
> This is incorrect. The original text read “where p is a 16-element vector of confidence scores derived from the model’s output logits (typically done using the softmax function)”, which clearly states that the probabilities are derived from logits (in the context of using the softmax function). We do not state that the logits are equivalent to probabilities.
>
> - _P7 line3 from bottom. "report both macro and micro AUC". It is unclear how the "macro AUC" is computed considering that the decision rule of ood need to take the maximum over all classes._
>
> Macro AUC is computed by computing the AUC for each class, where the samples used in the computation of each class’s AUC score are those samples where the model predicted the sample as that class. Then, each class’s AUC is averaged to yield the macro AUC score. We have added a description of macro AUC in the updated version of our paper.
>
> - _P5 line3 from bottom. It is not wise to pollute the raw data with simple augmentations, because (a) augmentations can be also used in training, which makes the perturbation of testing images pointless (b) augmentation creates simple artifacts that can be exploited by ood algorithms, making the dataset less useful._
>
> Importantly, we do not we apply augmentations to our final published dataset, so the the claim that the dataset is "less useful" due to augmentations is incorrect. We use augmentations to explore whether the models can solve the OOD task by merely looking for scanner-like noise (as a large portion of our OOD data is noise-free, unlike the original RVL-CDIP dataset. Our evaluation found that adding the augmentations to our test sets yielded little difference. Our use of augmentations is not part of our main findings and is not used in training. Regardless, augmentations have been used in prior work like the ImageNet-C dataset (https://arxiv.org/pdf/1903.12261.pdf; published in ICLR).
>
> - _only one ood (MSP) algorithm is used in ood evaluation, so the conclusion "struggle on the task of out-of-domain document classification" is less convincing._
>
> We have added an additional confidence score method that uses the Energy score from https://proceedings.neurips.cc/paper/2020/file/f5496252609c43eb8a3d147ab9b9c006-Paper.pdf, but the core finding that the models struggle on the out-of-domain document classification task remains the same. Additionally, in our original draft, we computed AUC on the RVL-CDIP test set versus RVL-CDIP-O, but in our updated draft we add analysis on AUC on RVL-CDIP-N versus RVL-CDIP-O, which is a more realistic and challenging setting (see Table 4 of the updated paper draft). Here we find that all models experience difficulty on the out-of-domain task.
>
> - _Not sure if it is manually labelled, or what the quality of labelling is._
>
> While the original draft mentioned details on how the dataset was collected/labelled, we added additional details and figures/tables in Section 3.2 of the updated draft.

---

> ### Comment · Reviewer_1zH1 · 2022-09-03
> **Thanks for the clarification**
>
> The concerns has been addressed in the revision. I have raised the score.

---

### Official Review · Reviewer_Z3qQ · 2022-07-27
**Two kinds of document datasets for evaluating performance on document classifiers.**

**Rating:** 6
**Confidence:** 4

**Strengths:**

1. This paper proposes a new and out-of-distribution dataset for the classification of documents.
2. They analyze the original RVL-CDIP dataset and point out its limitation. They also provide thorough reasons to show their dataset value.
3. Provide complete experiments and analysis.

**Weaknesses:**

1. The paper collects the OOD files by searching the categories which are out of 16 RVL-CDIP categories, but they do not provide which categories they use.
2. They may need to explain how they familiarize themselves with the 16 in-domain categories. Besides, one document may belong to two categories. The authors may need to show their filter process.
3. It would be great if they could hire human annotators to confirm the effectiveness of this dataset.

**Additional Feedback:**

Actually, I think the authors can publish a new dataset for document classifiers without accommodating to RVL-CDIP dataset. They can define their own categories and publish "born digital" documents.

It would be great if they can release their evaluation code.

**Clarity:**

The paper is well-written and easy to follow.

In figure 5, the authors may need to add the vertical axis label.

**Correctness:**

The evaluation procedures appear to be quite thorough. The results are also very well described and convincing. Only the data generation process may need more explanation for its filter process and the choice of categories.

**Documentation:**

The authors do not provide supplementary material. They only provide the data license on the last page of the main paper. It would be great if they could give more details about data usage and data maintenance. I may change my rating after they finish it.

**Ethics:**

The authors collect the pdf files from the search engine. But I am concerned that these pdf files may have copyright issues even though they make a statement on the last page. It would be great if an expert could review it.

**Relation To Prior Work:**

This paper does a good job explaining RVL-CDIP document classification and out-of-Distribution benchmark. Good summary.

**Summary And Contributions:**

The authors present out-of-domain and distribution shift datasets for document classification. They claim that the original RVL-CDIP corpus only includes 16 document categories and focuses on the American tobacco industry, which can not evaluate prior classifiers’ ability to handle out-of-distribution inputs. They collect their new categories of pdf files from search engines and the DocumentCloud repository. They also do data processing to make their new files fit the RVL-CDIP corpus. Based on their experiments, they show that both CNN architectures and image transformers exhibit accuracy drops on their dataset. This facilitates future research on document classification on out-of-distribution documents.

---

> ### Author Response · Authors · 2022-08-25
> **Response to Reviewer Z3qQ**
>
> Thank you for your review.
>
> - _The paper collects the OOD files by searching the categories which are out of 16 RVL-CDIP categories, but they do not provide which categories they use._
>
> This information was listed in Table 4 in the appendix of our original draft, but we moved it to Table 1 in the main text of our updated version of the paper.
>
> - _They may need to explain how they familiarize themselves with the 16 in-domain categories. Besides, one document may belong to two categories. The authors may need to show their filter process._
>
> We updated Section 3.2 to provide more discussion of the data collection process. In a nutshell: the original RVL-CDIP corpus does not have category descriptions, so we had to sample data from each category in order to familiarize ourselves with the types of documents in each category. Then, we proceeded with data collection for RVL-CDIP-N and RVL-CDIP-O. In particular for RVL-CDIP-N, we included only documents that were clear-cut examples of the RVL-CDIP target category. Appendix B shows examples from RVL-CDIP compared to RVL-CDIP-N.
>
> - _The authors do not provide supplementary material. They only provide the data license on the last page of the main paper. It would be great if they could give more details about data usage and data maintenance. I may change my rating after they finish it_
>
> The OpenReview platform may have made the supplementary material hard to find, but it was included in our original submission. In addition, there is a link to the dataset repository in the abstract of our paper. We provide it here: https://tinyurl.com/4he6my23.
>
> - _It would be great if they can release their evaluation code._
>
> We have updated the dataset repository to include evaluation code.

---

> > ### Comment · Reviewer_Z3qQ · 2022-08-27
> > **Thank you for your response**
> >
> > Thank you for solving my concerns. I have no further comments and raise my score.

---

> ### Author Response · Authors · 2022-08-25
> **ethics review**
>
> Please note that the official ethics review found that the "authors argument for 'fair use' is ok and i don't see a major ethical issue here".

---

### Official Review · Reviewer_5g94 · 2022-07-27
**Relevant enhancement for Document Image Classifiers**

**Rating:** 7
**Confidence:** 4
**Clarity:** The paper is well written and underst…

**Strengths:**

This new dataset allows to bechmark trained model for biases toward the training data. As highlighted in the paper, the state-of-the-art RVL-CDIP dataset is known to have some content bias, because of its origin (mostly related to tobacco industries, and documents older than 2006).

Thanks to this contribution, model developers will be able to better estimate their quality.

**Weaknesses:**

The dataset is still an order of magnitude smaller than the RVL-CDIP dataset, so it cannot be used for standalone training (which is anyway not the proposed purpose).

The data was acquired mostly via web searches, which would keep a proper licensing (for any purpose) unfeasible. The license states that the users of the dataset are responsible for acquiring proper licensing from the copyright holders for non-academic uses.

**Additional Feedback:**

If the authors defined some annotation guidelines, it would be nice to include them as supplementary material or in the dataset link.

**Correctness:**

The benchmarks and claims are correct and clearly reproducible.

It must anyway be noted that the authors "reversed engineered" the original classification of the RVL-CDIP dataset, which could have caused in some inconsistencies compared to the original annotation guidelines. This is an aspect that the authors could improve in the paper, answering for example questions like "which rules were used to classify the new documents?", "could one quantify/measure a precision metrics for comparing to the original classification?".

**Documentation:**

Authors claim in the checklist that benchmarks details (code, parameters, etc) are made available in the supplementary material, which unfortunately doesn't seem to be available for review in OpenReview.

The dataset itself it properly linked, and the reviewer was able to download and inspect it.

However, the structure of the dataset is not documented anywhere. It is pretty clear that the class is encoded in the filename, but a small documentation file would still be advised.

**Ethics:**

The reviewer agrees with the author's claim in the checklist. There is *no* need for a dedicated ethical review.

**Relation To Prior Work:**

The authors clearly describe how their work is complementary to existing state-of-the-art datasets. Benchmarks are also referencing the previous work properly.

**Summary And Contributions:**

The authors are enhancing the RVL-CDIP dataset with new out-of-distribution data. The authors consider both the out-of-domain (packaged in the new RVL-CDIP-N dataset) and the "new in-domain data" (packaged in the the RVL-CDIP-O dataset) aspect of the problem.

The paper is well written, and covers both the data acquisition and explains the usage of the new datasets with some relevant benchmarks.
As shown in the results, models trained on the current state-of-the-art RVL-CDIP dataset have a clear lower accuracy when tested on the new out-of-distribution data.

---

> ### Author Response · Authors · 2022-08-25
> **Response to Reviewer 5g94**
>
> Thank you for your review.
>
> - _It must anyway be noted that the authors "reversed engineered" the original classification of the RVL-CDIP dataset, which could have caused in some inconsistencies compared to the original annotation guidelines._
>
> We have updated the paper to include some additional discussion on how the new RVL-CDIP-O and RVL-CDIP-N data was gathered to address this. Additionally, Appendix B has been added to show examples of documents from RVL-CDIP-N compared to RVL-CDIP so that the reader can see the similarities between the two.
>
> - _Authors claim in the checklist that benchmarks details (code, parameters, etc) are made available in the supplementary material, which unfortunately doesn't seem to be available for review in OpenReview._
>
> We have updated the dataset repository to include more supplementary material, like training/inference code, parameters, model output, etc.
>
> - _However, the structure of the dataset is not documented anywhere. It is pretty clear that the class is encoded in the filename, but a small documentation file would still be advised._
>
> We have updated the dataset repository to include a README with documentation.

---

### Official Review · Reviewer_SHbs · 2022-07-27
**This paper proposes an interesting benchmark to evaluate the performance of document classification on out of distribution samples, which is a very common problem in industrial applications. The benchmark seems solid, although I am not fully convinced by the chosen metrics. The benchmark is run on many existing models. I would have liked seeing more insights as to why these models fail, and some suggestions on how to improve them (potentially with simple baselines).**

**Rating:** 7
**Confidence:** 4

**Strengths:**

1. This paper addresses a very interesting problem, that is relevant in practical (industrial) settings.
2. The proposed dataset covers two types of out-of-distribution samples, that are both very common in practical use cases.
3. Generated out-of-distribution samples have been carefully designed not to be too obvious to distinguish from real samples.
4. The evaluation metrics confirm the need of finer evaluation of classification models by showing that existing classifiers fail in the proposed setting.
5. Numerical experiments are rather extensive, and show consistent results on multiple models.

**Weaknesses:**

1. In ODD samples, some classes are highly underrepresented. As authors mention, finding examples in this class may be difficult, but I am worried this could eventually lead to mis-use of this dataset as a training+validation set of data for handling OOD data.
2. Metrics are not fully convincing to me. I would like a more "down-to-earth" measure, for instance the true positive ratio for a fixed false positive ratio of 1%/10%/50%. I believe this would be more meaningful for industrial applications (which the benchmark seems to target).
3. The author's benchmark shows that existing methods indeed fail on the OOD data. But authors do not provide insights as to why this happens. They only mention that training networks to specifically recognize out-of-domain could help, but do not provide a (even very simple) method to serve as a baseline for future work.
4. I am afraid that making a classifier recognize out-of-domain samples may also make the classifier mistake out-of-distribution samples as out-of-domain. Is this a potential issue? I would have liked some comment on the link between these two types of OOD samples.
5. Authors report some examples of misclassified OOD samples in Figure 6. While this makes a good visual representation, this does not give insights on what are the real challenges: what are the classes that are often mixed together? I imagine giving a confusion matrix could help on building intuition.
6. OOD samples are not obviously distinguishible from real ones, but the "scan-like" generation procedure still looks quite artificial and could possibly be identified by a classifier.
7. Authors mention a third type of OOD data: corrupted data. But they do not include corrupted records without really motivating this decision.

**Additional Feedback:**

The benchmark addresses a very relevant question. I believe better evaluation metrics would make it even more impactful.

**Clarity:**

The paper is very clearly written, with insightful visualization of representative examples of the different OOD classes. Experimentsal results are clearly exposed.

**Correctness:**

The author's approach to benchmarking classification of OOD documents appears logical to me.

**Documentation:**

The data collection process may be a little too vaguely described. Otherwise, sources of data, preprocessing, licenses, are properly described.

**Ethics:**

I am no licensing specialist, but the fact that the dataset uses some public data that is unlicensed under "fair use" (Appendix A) may be a little borderline.

On some specific classes, such as handwritten or resume, fairness issues could happen and may be interesting to evaluate in future work.


**Relation To Prior Work:**

The authors mention existing benchmark on classification on OOD samples in different settings. While they discuss the specificities of each setting, it is not clear to which extent ideas are shared between the current paper and these other works. Multiple works on classifiers for classification of documents are properly cited.

**Summary And Contributions:**

This paper proposes a dataset to evaluate document classifiers on out of distribution samples. They build a database that contains documents that are either not part of any class (out of domain) or in an existing class but sampled from a different distribution (out of distribution). The authors then propose metrics to evaluate classifiers on these out-of-distribution datasets, and compute them on multiple classical models, highlighting their weakness in this setting.

---

> ### Author Response · Authors · 2022-08-25
> **Response to Reviewer SHbs**
>
> Thank you for your review.
>
> - _Metrics are not fully convincing to me. I would like a more "down-to-earth" measure, for instance the true positive ratio for a fixed false positive ratio of 1%/10%/50%. I believe this would be more meaningful for industrial applications (which the benchmark seems to target)._ and _Authors report some examples of misclassified OOD samples in Figure 6. While this makes a good visual representation, this does not give insights on what are the real challenges: what are the classes that are often mixed together? I imagine giving a confusion matrix could help on building intuition._
>
> We have updated the paper to include more results including analyses on confidence threshold wrt accuracy, as well as confusion matrices (Tables 9-13). These can be found in Appendix A.
>
> - _I am afraid that making a classifier recognize out-of-domain samples may also make the classifier mistake out-of-distribution samples as out-of-domain. Is this a potential issue? I would have liked some comment on the link between these two types of OOD samples._
>
> This is indeed an issue, and our work highlights this important problem. Detectors aimed at classifying between in-scope (or in-domain) versus out-of-scope (out-of-domain) documents have tendency to mistake out-of-distribution documents as out-of-domain, even if they are in-domain (we see this with the RVL-CDIP-N dataset). We have included additional figures/analyses to highlight this problem. However, we do not see this as a limitation of our work; instead, our work highlights this problem and offers a new dataset for this task.
>
> - _OOD samples are not obviously distinguishible from real ones, but the "scan-like" generation procedure still looks quite artificial and could possibly be identified by a classifier._
>
> The artificial augmentations did not yield a clear pattern as to whether they were easier or harder than images that did not have the augmentations. Nevertheless, this is not a main experimental result in our paper.
>
> - _Authors mention a third type of OOD data: corrupted data. But they do not include corrupted records without really motivating this decision._
>
> We mentioned corrupted data in the Related Work section to give a more complete picture of landscape of the type of data that can be used as out-of-distribution for testing machine learning models. While we did conduct minor experiments using the Augraphy tool to introduce scanner-like noise to our documents, we did not find a meaningful relationship between presence/absence of these augmentations and model performance.
>
> - _I am no licensing specialist, but the fact that the dataset uses some public data that is unlicensed under "fair use" (Appendix A) may be a little borderline._
>
> Please see the official ethics review, which found that the "authors argument for 'fair use' is ok and i don't see a major ethical issue here".

---

> > ### Comment · Reviewer_SHbs · 2022-08-29
> > **Response to authors**
> >
> > Thank you for your response and for the additional figures. I agree that the proposed dataset indeed allows the evaluation of classifiers on the interaction between the two types of errors. I like the confusion matrices and think they can help in building intuition. I acknowledge the official ethics review on the fair use.
> >
> > I decide to keep my rating of 7.

---

### Official Review · Reviewer_JaJY · 2022-07-27
**Out-of-Distribution Performance on Document Classifiers**

**Rating:** 7
**Confidence:** 4
**Correctness:** All the claims are correct.
**Clarity:** The quality of presentation is high.

**Strengths:**

* the authors prepared a very useful addendum to the RVL-CDIP to
  measure the quality of classifiers when the prediction is done on
  out-of-distribution documents/classes (frankly, this should have
  been done by the Document Understanding community long time ago…)
* the paper presents results for a wide range of types of classifiers
  on the data sets (and showing a significant drop in quality when
  compared to the in-distribution test sets)


**Weaknesses:**

* the paper is not that innovative (still, the work is quite valuable
  for the community)
* using synthetic scanner noise (3.2, it would be better to find real
  scanned/non-born-digital PDFs)


**Additional Feedback:**

Minor fixes:

* "oon the full RVL-CDIP" => "on the full RVL-CDIP"

* "LayoutLM-v2" => "LayoutLMv2" (that's the way it is spelled in
  https://arxiv.org/abs/2012.14740v1)

* "may have consist" => "may have consisted"


**Documentation:**

Are the metadata for the PDFs obtained from the Internet available (at
least URLs)? Apart from that, I have no comments.


**Ethics:**

No issues found.

**Relation To Prior Work:**

Relation to prior work is clearly given.

**Summary And Contributions:**

The paper introduces an out-of-distribution "appendix" to the
well-known TVL-CDIP benchmark along with results obtained with a
number of state-of-the-art models.

---

> ### Author Response · Authors · 2022-08-25
> **Response to Reviewer JaJY**
>
> Thank you for your review.
>
> - _the paper is not that innovative (still, the work is quite valuable for the community_
>
> Our paper highlights the challenge of document classification in the presence of two types of out-of-distribution documents, a problem that is currently ignored by the RVL-CDIP-centric document understanding research sphere. Our results show that model performance suffers quite severely on our new out-of-distribution datasets compared to the standard RVL-CDIP test set, and therefore we believe it is a good fit for the Datasets & Benchmarks track at this conference as it points out a major gap in this research space. We agree with your earlier comment that a dataset like ours ought to have been introduced to the Document Understanding community earlier. In our experience, introducing a dataset like this can have a big impact to a specific research community.
>
> - _using synthetic scanner noise (3.2, it would be better to find real scanned/non-born-digital PDFs)_
>
> We agree, but our use of the synthetic scanner noise (via Augraphy) is not a main experiment in the paper, nor does our final version of RVL-CDIP-O and RVL-CDIP-N include synthetic scanner noise. We added more discussion on the use of Augraphy in the updated version of our paper.
>
> - _Are the metadata for the PDFs obtained from the Internet available (at least URLs)?_
>
> We include metadata like source (documentcloud vs. websearch), but not source URL, as URLs are prone to being  changed or made obsolete by each website so that the original URLs are no longer valid.
>
> - _Minor fixes_
>
> Thank you for pointing these out; these have been fixed and we will upload an updated draft shortly.

---

### Review · Ethics_Reviewer_YLRS · 2022-08-23

**Recommendation:** 1

**Ethics Review:**

2 over 5 reviewers flagged the paper with ethics issue related to licensing as  the dataset uses some public data that is unlicensed under "fair use" (Appendix A) ... but reading appendix A i think authors argument for 'fair use' is ok and i don't see a major ethical issue here

---

### Meta-Review · Area_Chair_8Agz · 2022-09-07

**Recommendation:** Accept
**Confidence:** 5

**Metareview:**

This paper focus on evaluating dataset bias (out-of-distribution), particularly on document image classier. All reviewers find the evaluation technically sound thorough and insightful. All of them recommend acceptances. The authors have carefully addressed all reviewers concerns. Therefore this paper is suitable for accept.

---

### Decision · Program_Chairs · 2022-09-16

Accept